# Microstructure and Hardness of Spark Plasma Sintered Inconel 625-NbC Composites for High-Temperature Applications

**DOI:** 10.3390/ma14164606

**Published:** 2021-08-16

**Authors:** Adrian Graboś, Jan Huebner, Paweł Rutkowski, Shenghua Zhang, Yen-Ling Kuo, Dariusz Kata, Shigenari Hayashi

**Affiliations:** 1Faculty of Materials Engineering and Ceramics, AGH University of Science and Technology, 30-059 Cracow, Poland; huebnerj@agh.edu.pl (J.H.); pawelr@agh.edu.pl (P.R.); kata@agh.edu.pl (D.K.); 2Faculty of Engineering, Hokkaido University, Sapporo 060-8628, Japan; shenghzhang_989@163.com (S.Z.); yenling.kuokid@gmail.com (Y.-L.K.); hayashi@eng.hokudai.ac.jp (S.H.)

**Keywords:** Inconel 625, niobium carbide, spark plasma sintering, metal matrix composites

## Abstract

The study focuses on obtaining Inconel 625-NbC composites for high-temperature applications, e.g., jet engines, waste-to-energy combusting systems or gas engine turbines, and characterizing them in terms of their microstructure and hardness improvement. Synthesis was performed utilizing Spark Plasma Sintering (SPS) at 1150 °C under the load of 45 MPa in medium vacuum (under 10^−3^ MPa) for a total time of 60 min. Four sets of samples with different Inconel 625 to NbC weight ratios were prepared (5, 10, 20, and 30 wt.%), followed by a reference sample containing no ceramic reinforcement. Obtained materials were hot-rolled at 1150 °C with a 10% reduction step and later cut and polished to perform characterization utilizing scanning electron microscopy (SEM) equipped with energy dispersive spectroscopy (EDS) module and microhardness testing device equipped with Vickers indenter. Hardness was improved proportionally to NbC addition achieving an increase of up to 20% of reference values. Additional heat treatment was conducted on the hot-rolled samples at 1200 °C in an argon atmosphere to further observe the interaction between reinforcement and alloy. Their microstructure revealed the coarsening of precipitates within the metal matrix and partial reinforcement dissolution, which proved to be crucial to obtaining the highest quality composites with homogenous hardness improvement.

## 1. Introduction

An Ni-based superalloy, Inconel 625, is an austenitic material that possesses advantageous properties for high-temperature applications, such as gas turbines, jet engines, waste-to-energy combustions, or metal arc-welding. This alloy is easily weldable, tough, ductile, highly resistant to wear or erosion/corrosion, and suitable for work at elevated temperatures [1,2,3,4]. Progress in their development is intertwined with higher work temperatures and/or higher mechanical strength. Therefore, to meet constantly increasing standards and emerging working conditions, while simultaneously prolonging the life of exposed elements, new materials are being developed. In terms of improving mechanical performance, metal matrix composites (MMC) are one of the most promising solutions [5,6,7].

When ceramic reinforcement is introduced to the composite, improvement of certain mechanical properties is expected to occur, e.g., hardness. A similar effect is observed in MMC materials with the additional formation of secondary oxides or carbides, created due to their reaction with alloying elements. The exact character of the new material synthesis depends on various parameters. That being said, ceramic addition into a metal matrix generally follows a simple pattern. Properties typical to ceramics increase proportionally to the amount of addition, provided that a critical volume of reinforcement is achieved. Then, after reaching a certain amount (different for each material), a maximum is reached and further increase of additives works detrimentally. This amount is often referred to as a “peak” value for the composite properties. Observed detrimental effects occur due to introduced processing difficulties, such as additional porosity, increased brittleness, a surplus of certain elements in the material, etc. Such maximum can be shifted by optimizing process parameters, typically process time and/or temperature.

During the prolonged high-temperature exposition of Inconel 625 superalloy, precipitates may appear in the matrix. Crystal structures of those precipitates are taking the form of topologically close-packed (TCP) phases consisting of δ, µ, P, and Laves phases and carbides. There are many pieces of research [8,9,10] about the character, influence, and conditions of their presence in various compounds. It is generally agreed upon that TCP precipitations are undesirable, solely due to the significant depletion of alloying elements from the matrix. In addition, as brittle, intermetallic inclusions, TCP phases are a cause for the reduced ductility of alloy with accompanying pore formation, possibly reducing fracture toughness or accelerating crack propagation. While through the assumption of mixture law, individual hardness of TCP phases could overall improve this property in the composite, it is yet to be proven that it can be implemented industrially.

It has been already reported in previous research articles that introducing carbides to Inconel 625 alloys as a form of MMC does have the potential to improve those materials in terms of their mechanical properties and their general utility for high-temperature applications. For example, Cao et al. noted the solid solution strengthening of superalloys employing nano-sized TiC in laser melting deposition and utilizing a high cooling rate during their process [11]. Huebner et al. obtained an Inconel 625-WC system with homogeneously distributed reinforcement that increased the overall hardness of the material, noting the influence of preserved TCP phases during microhardness tests. It was summarized that the addition of carbides was resulting in an increasing amount of TCP phases observed in the microstructure, usually appearing after extensive heat treatment [12]. The conclusion can be drawn that such reinforcement does improve alloy performance under increased mechanical stress with the downside of TCP phase formation, accompanied by previously discussed detrimental effects.

Simultaneously, it has been reported that the addition of Nb aided the formation of the secondary γ″-Ni_3_Nb phase, which is regarded as a strengthening phase in Ni-based superalloys. This is however reduced by two factors: additional elements in the mixture and Fe presence [13]. In previously mentioned Huebner’s work, it was discovered that higher amounts of WC led to the extensive formation of secondary phases that provided additional “anchoring” of fine carbide particles in an Ni-based matrix. It was concluded that it was beneficial for the overall hardness of the material in comparison to composites with lower amounts of ceramic reinforcement. This leads to the question if introducing carbide in the form of NbC would be a suitable solution to this problem. As ceramic reinforcement with promising mechanical properties, it should work similarly to WC or TiC, without introducing additional elements to the alloy mixture and thus, possibly reducing the formation of TCP phases.

The aim of this study was to obtain and evaluate the Inconel 625-NbC composite. Reducing the amount of TCP phases in the material can be achieved in two possible ways: by controlling the process of solidification, specifically by reducing its time [14], or by introducing additional elements (e.g., Ru), that suppress creation of those phases to the alloy mixture [15]. Therefore, Spark Plasma Sintering (SPS) was utilized as the main manufacturing method in this process, as it allows the synthesis of bulk composite bodies with short exposure to high temperatures. Additionally, materials produced by the SPS method are reported to be of near theoretical density for simple compounds. This allows us to expect that it will be a countermeasure to porous areas typically created during the insertion of ceramic phase into metal matrix [16,17].

## 2. Materials and Methods

For the purpose of composite synthesis, the following commercial reactants were used: 99.0% purity NbC powder produced by Japan Metal Service and 99.0% purity Inconel 625 provided by New Metals and Chemicals Corp. Ltd. (Tokyo, Japan). Powder size distribution was measured by Microtrac-Bell MT3300 EX II (Microtrac, Osaka, Japan) laser diffraction analyzer. Hardness tests were performed under a load of 200 g for 15 s per measurement, utilizing Future-Tech FM-700 (Future-Tech, Gliwice, Poland) equipped with Vickers indenter. Microstructure of the samples was observed by Scanning Electron Microscopy (SEM) with energy dispersive spectroscopy (EDS) microanalyzer in JEOL JSM-6390 (Jeol, Peabody, MA, USA) and NOVA NANO SEM 200 (FEI, Eindhoven, The Netherlands) microscopes. X-ray diffraction (XRD) analysis was performed utilizing a PANalytical X-ray diffractor (Malvern Panalytical, Almelo, The Netherlands), equipped with a Cu tube. XRD results were analyzed using X-pert HighScore software. Image analysis was conducted with ImageJ software.

Figure 1 presents the scheme of core procedures for the obtaining and characterization of Inconel 625-NbC MMC materials.

Commercially available powder of Inconel 625 characterized in Table 1 was mixed with NbC powder to obtain 5 sets of samples. Phase composition of samples was listed in Table 2. Pure Inconel 625 polycrystals were also prepared by SPS as reference samples for experimental work.

The particle size distribution of the NbC powder was ranging from 0.1 to 80 µm, as shown in Figure 2. The powder had a bi-modal character of distribution. Powders with a grain size below 1 µm visible as the first maximum were approximately a third of the total sample volume, 95% of the grains were of a size below 10 µm, and the remaining 5% ranged up to a noted maximum of 74 µm. Powder mixtures were dry homogenized using a rotary mill with steel milling balls as a grinding media for 24 h. The obtained material was placed in a cylindrically-shaped, graphite die with 3 layers of isolation: carbon sheet, aluminum oxide fabric, and nickel sheet lined internally, as shown in Figure 3.

A three-layer form of isolation prevented the reaction between the sample and graphite die, while simultaneously maintaining thermal stability during the whole process and enclosing sample for further processing. The protection of the samples was removed by mechanical grinding prior to further steps.

Spark Plasma Sintering was conducted in a medium vacuum kept under an atmosphere of 10^−3^ MPa to prevent oxidation during the reaction. Process parameters were collected in Table 3.

A bulk composite obtained by SPS was placed in a steel form with aluminum oxide powder as an isolation medium to conduct the first stage of hot-rolling at 1150 °C. The hot-rolling process was involved to obtain thin plates with uniform thickness for further testing and to initiate the recrystallization of alloy, which was expected to reduce the amount of porosity and other defects within the substrate. The reduction rate for each rolling pass was set to 10% and the final thickness of obtained samples equaled approximately 5.0 mm. Such samples were used for microstructural observations and hardness tests. Both studies were repeated after those samples were annealed at 1200 °C for 24 h.

To determine the hardness of the obtained polycrystals, Vickers method was used. All materials were tested under the load of 200 g (approximately 1961 mN) for 15 s. Both standardly obtained and heat-treated samples were characterized. Each sample was measured at least 10 times to calculate the average hardness through the whole material. Utilizing a microhardness testing device allowed for the differentiation between the matrix and reinforcement.

## 3. Results

### 3.1. Microstructure Observation

Microstructure appearance clearly revealed the presence of the NbC reinforcement phase within Inconel 625 superalloy matrix, as shown in Figure 4, and was later qualitatively confirmed by EDS analysis. Reinforcement was directly distributed in the whole substrate of material, supposedly along the Prior Particle Boundary (PPB), which occurs in Ni-based alloys during high-temperature/high-pressure processes, in this case, SPS followed by hot-rolling. Larger quantities of NbC resulted in a more uniform distribution along the PPB, which is shown in Figure 4c,d. Smaller additions of NbC (Figure 4b) displayed a tendency to form locally concentrated inclusions. Based on the obtained microsections, the area fraction of porosity and an average size of pore were estimated by micrograph analysis (Table 4). It can be noted that the amount and size of pores decreased significantly for 5 and 10 wt.% additions of NbC in comparison to the NbC-free Inconel 625 reference sample obtained by the same consolidation process. The grain size difference between the NbC and Inconel matrix could allow for reducing pore formation during the consolidation process by filling smaller particles in the gap between bigger particles. However, the sample with the addition of 20 wt.% of NbC had displayed multiple occurrences of different porosity instances present near reinforcement particles. The amount of NbC is high enough to introduce multiple contacts between reinforcement particles, which resulted in the formation of additional pores. This suggests that above this threshold, the sintering process becomes complicated for the Inconel 625-NbC system with the current consolidation condition and powders used in the present study. It was also supported by the fact that samples with a 30% addition of NbC were too brittle to characterize, due to the loss of their integrity. Thus, samples with an NbC amount of above 20 wt.% were not evaluated by any further test.

Careful examination of the obtained samples revealed the character of NbC reinforcement inclusions within the material (Figure 5). It can be observed that, in fact, every amount of NbC addition is resulting in visible pores along the PPB and between NbC reinforcement particles. Presented samples differentiated in the concentration of mentioned microstructural defects. The 625_NbC_5 sample had fewer pores and cracks, thus, the reinforcement-matrix has better contact within the samples. Whereas the 625_NbC_20 contained greater defect distribution existing parallel to the reinforcement in the matrix. That suggests a significant decrease in the mechanical properties of the 625_NbC_20 due to overall brittleness when compared to the rest of obtained samples. The greater amount of defect formation could occur due to the formation of large secondary precipitates in the Inconel 625 matrix, since precipitate density was higher for the composite with higher NbC addition, as will be discussed later. Better homogenization of the sample would probably reduce the number of defects present near reinforcement in any samples.

Combining SEM images with EDS analysis allowed for a qualitative examination of observed microstructures and their results are presented below (Figure 6).

Observation confirms that NbC reinforcement is well embedded in the matrix along the PPB, indicating that the synthesis product was properly established as an MMC material. EDS results allow for identifying precipitate with bright contrast within the γ-Ni matrix. It was confirmed to be Nb/Mo-rich phase and their presence was also confirmed within the sample without reinforcement present. However, the Cr and Mo contents in Nb/Mo-rich phase formed in MMC were found to be higher than that formed in the sample without NbC reinforcement. The composition in the γ-Ni matrix was comparable for both samples.

The metastable γ″ and/or stable δ phases (e.g., Ni_3_Nb or Ni_3_Mo), which increase mechanical properties of the alloy, can potentially exist in those samples [13]. Whilst γ″ are generally formed below 1000 °C, alloys obtained by SPS could contain mainly stable δ phase. However, those precipitates were not confirmed in this observation. At the same time, previously mentioned detrimental TCP phases co-existing in Ni-Cr-Mo or Ni-Cr-Nb systems were also not observed within those microstructures, even in higher magnification images.

It is worth to mention that NbC reinforcement does not interrupt the formation of Nb/Mo-rich phases and similar precipitates can be specified within each sample. As it appeared from Figure 5 and Figure 6, their quantity increased visibly with higher amounts of NbC added. Such a relation was a consequence of introducing additional Nb and C from the NbC reinforcement to the alloy matrix. Therefore, those compounds are likely to be carbides, which can be supported by the resulting higher Mo content within precipitates in samples with NbC additions. Exact EDS qualitative results were collected in Table 5 as confirmation. Carbon concentration was not included because EDS analysis is not accurate for C.

A detailed microstructure of reinforcement was taken to examine the reaction between the NbC and metal substrate as shown in Figure 7. A reaction layer with intermediate contrast was formed between the NbC and 625 matrix, although it was too thin to analyze by conventional SEM technique. However, most of the NbC was present as inclusion within material characterized by its bulky grains. This was significantly different from the typical eutectic structures with TCP phases in the Inconel 625-WC composite, presented in Huebner’s et al. reports covering similar materials. As a consequence, the previously mentioned “anchoring” effect is not occurring between the NbC and Inconel 625 matrix [12].

Figure 8 shows samples after additional heat treatment at 1200 °C. In those figures microstructures before heat treatment were also shown as a reference (right-side images). The additional heat treatment resulted in the coarsening of the Nb/Mo-rich precipitates in the matrix for both samples, with and without NbC reinforcement. The number of the precipitates was decreased after heat treatment due to coarsening. Moreover, the reinforcement along PPB tended to be discontinuous, particularly in the areas where the reinforcement was distributed thinly, suggesting that the reinforcement also reacted with the matrix during a second heat treatment at 1200 °C. This tendency was more clearly observed in the 625_NbC_5 sample.

The following EDS analysis (Table 6) revealed that Mo content in the Nb/Mo-rich precipitate further increased, but Nb content tended to decrease. The increase in Mo content was greater for the samples with NbC than without NbC and it can be noted that simultaneously Si tended to increase in the precipitates.

### 3.2. Hardness

Results of the microhardness tests were collected in Table 7. Overall, the addition of NbC resulted in increased hardness of Inconel 625 materials. The hardness increased with an increasing amount of NbC up to 10%, but further addition of NbC did not increase the hardness. Microhardness tests revealed that increased precipitate formation occurring due to NbC addition was beneficial for the final properties of the composite matrix, as its hardness was increased similarly to the whole material.

In order to confirm this conclusion, a micrograph analysis was conducted to compare the area fraction of Nb/Mo-rich precipitates, and results were collected in Table 8. It was observed that the area fraction of precipitates increased with the increasing amount of NbC. As discussed below, the coarsening of precipitates by a heat-treatment at 1200 °C was also confirmed. Notably, the area fraction of coarsened precipitates was slightly decreased with increasing amounts of NbC.

To determine how observed microstructural changes were affecting Inconel 625-NbC hardness, additional tests were performed. Results were collected in Table 9.

Annealing of the material allowed to further increase composite hardness. A significant and uniform increase of hardness can be observed for the heat-treated 625_NbC_5 sample. It was concluded that the obtained composite material, due to precipitate phase distribution (in the spectrum of all our samples) displays a beneficial strength profile.

### 3.3. X-ray Diffraction

To determine the character of phase transformations occurring in the obtained materials, a set of chosen samples was measured utilizing XRD analysis (Figure 9).

It was observed that SPS synthesis of the Inconel 625-NbC materials leads to microstructures composed of multiple phases, which were not easily distinguishable by diffraction method. An implemented heat-treatment process enabled a more stable phase composition. Obtained quantifications of occurring phases matches with the analysis of area fraction of precipitates, as elements forming those compounds. From these diffraction patterns, it can be confirmed that Nb/Mo-rich phase is Mo_2_C type carbide.

## 4. Discussion

In the previous investigation [12], introducing WC in order to improve hardness and wear resistance into an Ni-based superalloy was reported to result in the formation of mechanically detrimental TCP phases. However, obtained Inconel 625-NbC composites were free of those intermetallic phases. Simultaneously, the number of precipitates within the matrix increased in correlation to the amount of added NbC. TCP formation did not occur even after additional heat treatment, which would prolong the exposure of the material to high temperatures. Instead, the coarsening of existing precipitates and compositional change of the Nb/Mo-rich precipitate were observed.

The increase of volume fraction of Nb/Mo-rich precipitates can be attributed to the partial dissolution of NbC reinforcement. Higher carbon supply from the NbC dissolution in the composite with higher NbC addition promotes the formation of higher Nb/Mo-rich carbide in the matrix.

Comparing the stability of carbides present in Ni-based superalloys and obtained XRD patterns allows discussing microstructural changes observed in obtained composites. After additional heat treatment at 1200 °C, a reduction of the area fraction of precipitates accompanied with their coarsening occurred. This might derive from a decrease in the molar volume of compounds due to the phase transformation of carbides from M_6_C and M_23_C_6_ carbides (M stands for metal element) to MC/M_2_C carbides. It was found that Mo content in the precipitate increased with the simultaneous decrease of Nb and Cr contents, which suggests the following phase transformation of carbides occurs during a heat treatment. Initially, complex M_6_C and M_23_C_6_ carbides and intermetallic compounds may have emerged after the SPS process [13,18]. However, those compounds are typically not preserved during a heat treatment at 1200 °C. They are dissolved similarly to intermetallic phases present in Inconel 625, and are leaving more stable MC/M_2_C carbides at higher temperatures, resulting in decreasing Cr content in the carbides [19,20]. This was also confirmed by XRD analysis.

Observed microstructural changes had influenced following hardness tests. Increased precipitation was resulting in an increase of alloy matrix hardness, which was important for the average hardness of the composite. Heat treatment of the samples proved to be potentially beneficial for this process. If certain phase distribution (as described in Section 3.2) is obtained, the composite can be decisively improved.

In the present study, the effect of reinforcement which was present along PPBs was not discussed since a lot of defects, such as pores and cracks, were introduced within it. A more homogeneous distribution of reinforcement within the matrix is necessary to assess the effect of reinforcement. Further study with different particle sizes of NbC and 625 powders is necessary to optimize the microstructure of reinforcement. This could improve the mechanical properties of MMC with NbC reinforcement.

## 5. Conclusions

Presented results can be concluded as follows:

Spark Plasma Sintering stands as a viable method of obtaining Inconel 625-NbC composites. Dense materials were produced up to a 20 wt.% of NbC. Higher amounts of carbide resulted in introducing a large amount of defects, such as pores and cracks, which increased their brittleness and resulted in an overall loss of integrity.

The introduction of NbC into the Inconel 625 alloy increased the area fraction of precipitates within the metal matrix, forming Nb/Mo-rich phases, which were considered secondary carbides. Additional heat-treatment leads to precipitate coarsening and reduction of their area fraction.The introduction of NbC into the Inconel 625 alloy creates the MMC with overall increased hardness as additional precipitation increases matrix hardness.The best performance of Inconel 625-NbC MMC was obtained combining an optimized ceramic amount with additional heat treatment, which led to unique secondary phase distribution in the microstructure. In this study, it was achieved for 625_NbC_5 samples.

## Figures and Tables

**Figure 1 materials-14-04606-f001:**
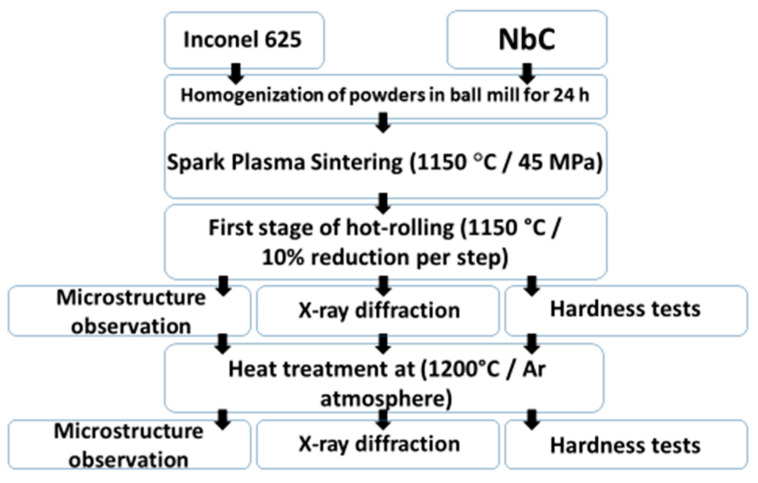
Schematic representation of the experimental procedure.

**Figure 2 materials-14-04606-f002:**
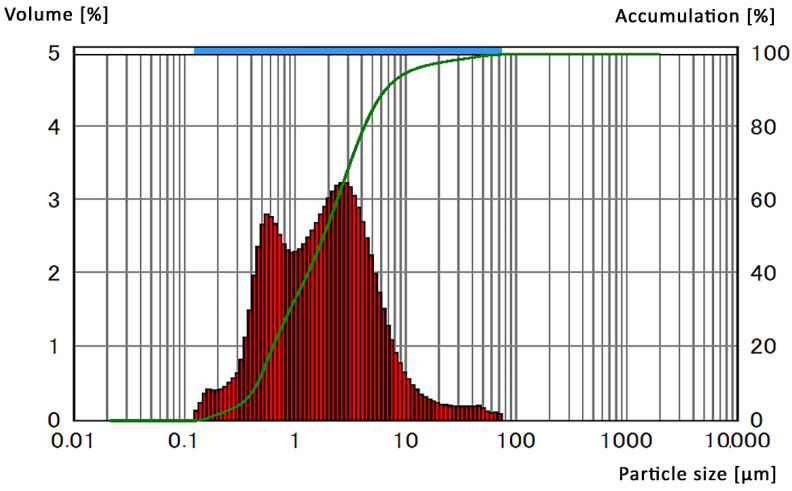
Powder size distribution of NbC measured by DLS method.

**Figure 3 materials-14-04606-f003:**
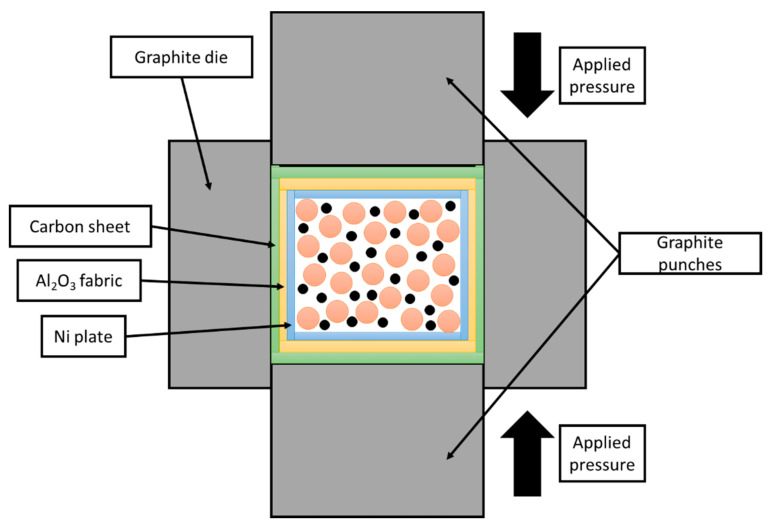
Sample emplacement for Spark Plasma Sintering.

**Figure 4 materials-14-04606-f004:**
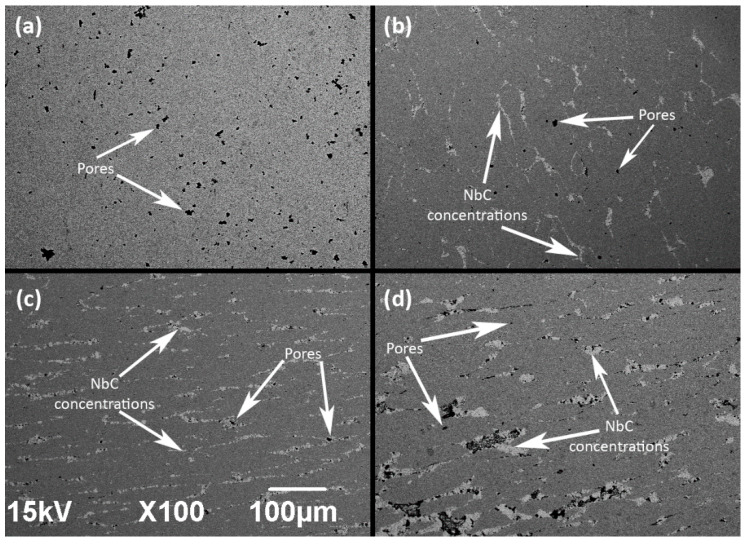
Microstructure of sample (**a**)—Pure Inconel 625 (reference sample), (**b**)—5% NbC, (**c**)—10% NbC, (**d**)—20% NbC reinforced Inconel 625 metal matrix after the hot-rolling. Scale applies to each micrograph.

**Figure 5 materials-14-04606-f005:**
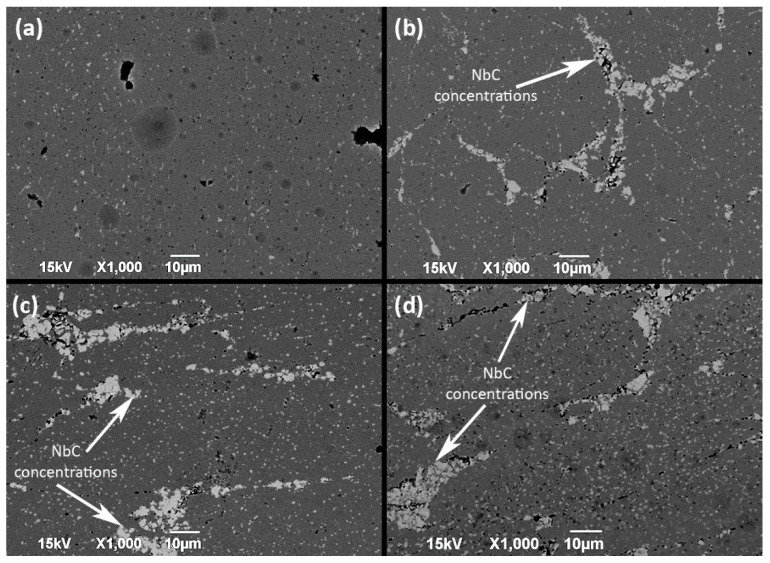
Enlarged microstructures of (**a**)—Pure Inconel 625 (reference sample), (**b**)—5% NbC, (**c**)—10% NbC, (**d**)—20% NbC reinforcement and its influence on microstructure of Inconel 625 MMC. Scale applies to each micrograph.

**Figure 6 materials-14-04606-f006:**
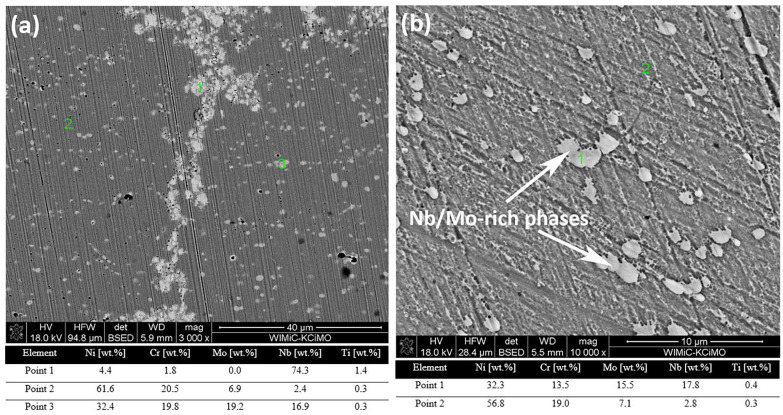
SEM images and EDS point analysis for samples of (**a**) 5% of reinforcement and (**b**) without reinforcement.

**Figure 7 materials-14-04606-f007:**
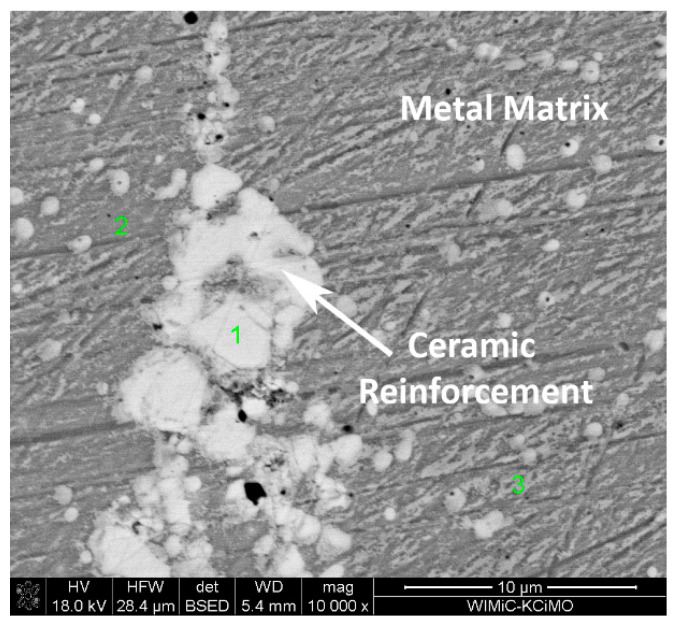
Example of interaction between ceramic reinforcement and metal matrix in obtained Inconel 625-5%NbC MMC materials.

**Figure 8 materials-14-04606-f008:**
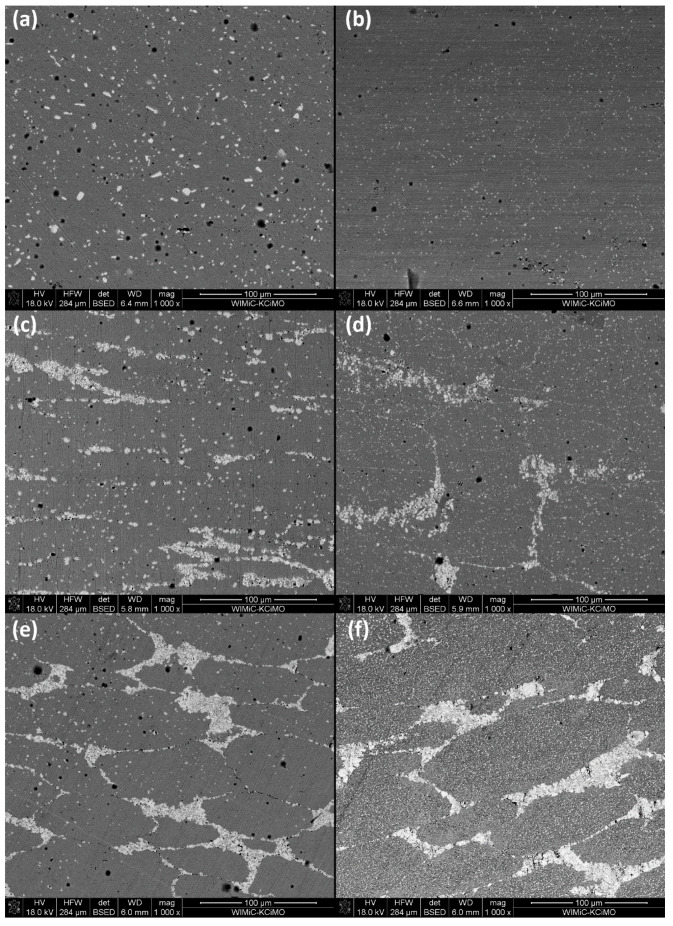
Comparison between heat-treated and reference samples. Compared samples: (**a**,**b**)—Pure Inconel 625 (reference sample); (**c**,**d**)—NbC 5%, (**e**,**f**)—NbC 20%. Left-sided materials were heat-treated, while right-sided images are matching composites observed before this process.

**Figure 9 materials-14-04606-f009:**
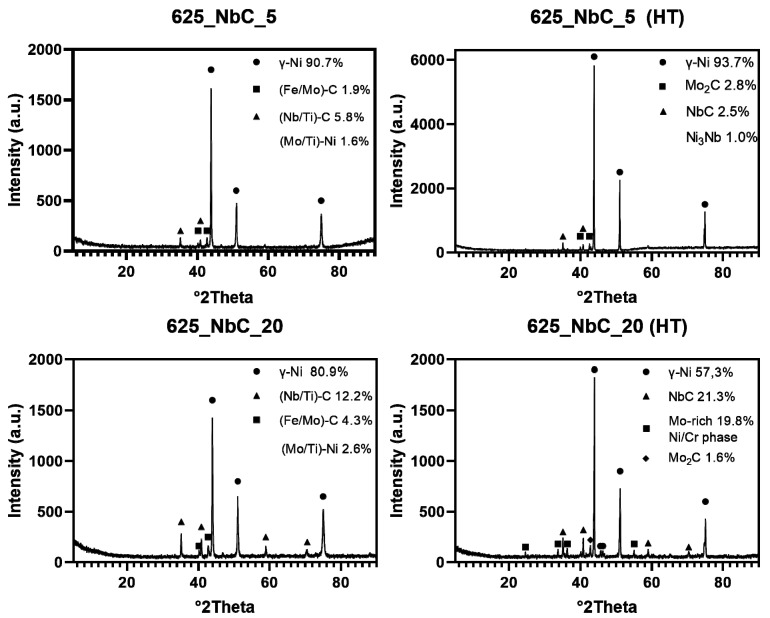
XRD patterns of chosen Inconel 625-NbC samples and phase quantification. HT stands for “heat-treated”. Phases without a symbol where found in the material without clear, separate peak by analysis process provided in the software.

**Table 1 materials-14-04606-t001:** Composition of provided Inconel 625 powder, where bal. stands for balance composition.

Element	Ni	Cr	Fe	Mo	Nb	C	Mn	Si	Al	Co	Ti
Amount [%]	bal.	20–23	<1.50	8–10	3.15–3.85	<0.03	0.2–0.5	0.3–0.5	<0.10	<1	<0.10

**Table 2 materials-14-04606-t002:** Composition of samples.

Sample Series	Inconel 625 [wt.%]	NbC [wt.%]
625_NbC_0	100	0
625_NbC_5	95	5
625_NbC_10	90	10
625_NbC_20	80	20
625_NbC_30	70	30

**Table 3 materials-14-04606-t003:** Process parameters of Inconel 625-NbC Spark Plasma Sintering.

**Time**	Heating	18 min
	Synthesis	10 min
	Total	60 min
**Pressure**	Load	45 MPa
	Atmosphere	<10^−3^ MPa
**Temperature**		1150 °C
**Average heating rate**		60 °C/min
**Sample diameter**		20 mm

**Table 4 materials-14-04606-t004:** Porosity estimations for obtained Inconel 625-NbC materials.

Sample	Porosity Area [%]	Average Size [µm]
625_NbC_0	0.883	10
625_NbC_5	0.219	6
625_NbC_10	0.261	4
625_NbC_20	0.758	6

**Table 5 materials-14-04606-t005:** EDS qualitative analysis of Inconel 625-NbC materials.

Sample	Phase Type	Ni wt.%	Cr wt.%	Mo wt.%	Nb wt.%	Si wt.%	Ti wt.%
625_NbC_0	Matrix	56.8	19.0	7.1	2.8	0.4	0.4
	Nb/Mo-rich phases	32.3	13.5	15.5	17.8	2.3	0.3
	Ceramic reinforcement	-	-	-	-	-	-
625_NbC_5	Matrix	61.6	20.5	6.9	2.4	0.5	0.3
	Nb/Mo-rich phases	32.4	19.8	19.2	16.9	0.6	0.2
	Ceramic reinforcement	4.4	1.8	0	74.3	0	1.4
625_NbC_10	Matrix	60.4	20.8	6.1	2.6	0.5	0.3
	Nb/Mo-rich phases	30.5	17.7	19.7	18.2	0.7	0.3
	Ceramic reinforcement	3.9	2.0	0	73.9	0	1.4
625_NbC_20	Matrix	61.1	20.0	6.0	2.4	0.7	0
	Nb/Mo-rich phases	30.0	16.4	20.9	18.0	1.0	0.4
	Ceramic reinforcement	3.6	1.6	0	74.2	0	1.5

**Table 6 materials-14-04606-t006:** EDS qualitative analysis of heat-treated Inconel 625-NbC materials.

Sample	Phase Type	Ni wt.%	Cr wt.%	Mo wt.%	Nb wt.%	Si wt.%	Ti wt.%
625_NbC_0	Matrix	67.3	21.5	8.2	1	0.3	0.1
	Nb/Mo-rich phases	39.0	25.8	18.9	14.3	0	0
	Ceramic reinforcement	-	-	-	-	-	-
625_NbC_5	Matrix	59.6	20.1	7.0	3.4	0.4	0.4
	Nb/Mo-rich phases	29.6	13.2	26.3	16.9	2.3	0.4
	Ceramic reinforcement	5.8	2.7	0	71.2	0	1.1
625_NbC_10	Matrix	65.8	22.2	7.0	3.4	0.4	0
	Nb/Mo-rich phases	29.8	17.2	24.0	13.9	1.8	0.4
	Ceramic reinforcement	3.0	1.2	0	78.0	1.7	0
625_NbC_20	Matrix	62.3	21.0	7.2	4.2	0.8	0.3
	Nb/Mo-rich phases	37.5	15.8	24.4	14.3	2.6	0.3
	Ceramic reinforcement	2.7	1.3	0	79.7	1.4	1.1

**Table 7 materials-14-04606-t007:** Hardness of Inconel 625-NbC materials.

Sample	Average Hardness with NbC Reinforcement [HV]	Matrix Hardness without NbC Reinforcement [HV]
625_NbC_0	323 ± 4	323 ± 4
625_NbC_5	339 ± 14	327 ± 4
625_NbC_10	358 ± 16	346 ± 5
625_NbC_20	360 ± 25	345 ± 6

**Table 8 materials-14-04606-t008:** Area fraction of precipitates in Inconel 625-NbC materials before and after heat treatment.

Sample		625_NbC_0	625_NbC_5	625_NbC_10	625_NbC_20
Area fraction of precipitates [%]	Standard	14	16	21	30
	Heat-treated	10	9	6	7

**Table 9 materials-14-04606-t009:** Hardness of heat-treated Inconel 625-NbC materials.

Sample	Average Hardness with NbC Reinforcement [HV]	Matrix Hardness without NbC Reinforcement [HV]
625_NbC_0	315 ± 10	315 ± 10
625_NbC_5	372 ± 10	366 ± 2
625_NbC_10	357 ± 21	346 ± 3
625_NbC_20	367 ± 26	348 ± 10

## Data Availability

Not applicable.

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
