# Peer review of "Microstructure and Hardness of Spark Plasma Sintered Inconel 625-NbC Composites for High-Temperature Applications"

_materials, 2021, doi:10.3390/ma14164606_

Round 1

Reviewer 1 Report

In this work inconel 625-NbC composites for high-temperature applications produced by ball milling and spark plasma sintering were investigated. The main remark to the paper is the absence of XRD data and making assumptions in the discussion about structural transformations and formation of preceptates without XRD data. It would be nice to confirm it by XRD.
Of the minor remarks, the following can be distinguished:
1. The section on materials and methods does not discribe the parameters of milling and what ball materials were used and how they could contaminate the final material, as well as this effect;
2. In the caption to Figure 5, it is not clear what Figure d refers to. In general, the figures are not very well arranged and confuse the reader. It is logical to arrange photographs of microstructures as the additives increase 0 - 5 - 10 - 20, and in the text of the paper of the image of structure without additives, it appears either at the beginning or at the end;
3. The paper indicates that the NbC particles agglomerate, but it can be seen that some of the particles pass into the matrix. It would be good to determine the fraction of particles in the matrix.
4. According to Tables 6 and 8, the hardness of the matrix without particles increases with increasing niobium carbide, but the reasons are not discussed.  
The paper and results is interesting for scientific community.

Reviewer 2 Report

The article is written on a relevant topic and is devoted to obtaining a composite material with enhanced properties.

It is especially worth noting the very good structure of the article, clear and competent compilation of research results.

At the same time, the article can be slightly improved if:

  1. In fig. 4 you can indicate the pores, it will be useful to the reader.
  2. The paper talks about porosity and its effect on properties, but a quantitative assessment was not carried out. You can add the results of determining the number of pores.
  3. 6 is of poor quality. Measurement points 1 and 3 are indistinguishable. The quality of the photo should be improved.
  4. Line 284 talks about using WC. This is mistake? If this is not a typo, what kind of research is mentioned? Provide a link.

Reviewer 3 Report

This manuscript reports the evolution of microstructure and hardness of Inconel 625-NbC composites processed by spark plasma sintering. The authors investigated how the microstructure and hardness changes with respect to the NbC content and post heat treatment. There are several issues needed to be addressed and revised. Please, see the below.

1) "... and/or higher mechanical stress." in introduction part. The authors should distinguish between stress and strength. 

2) The sentence "After a critical volume of reinforcement is added, desired 43 properties are increased proportionally to the amount of addition." in introduction part should be revised.

3) Last sentence of the second paragraph in introduction part should be revised.

4)  What is solid strengthening? Do the authors mean solid solution strengthening?

5) The authors should add spacing before using unit.

6) The authors claimed that reinforcement was homogeneously distributed in the whole substrate of material. However, they don't look homogeneously distributed at all. 

7) What is the mechanism behind the distribution of defects parallel to the reinforcement in the matrix? 

8) Caption in fig. 5 should be revised.

9) Recommend quantifying the fraction of NbC, Nb/Mo-rich phase. 

10) To make clear that the compounds are likely to be carbides, crystallographic analysis is needed.

11) What is the meaning of "...where the reinforcement was distributed with thinner thickness"? what is the thinner thickness?

12) What is the mechanism behind peak hardness value of the 625_NbC_5 among the heat-treated Inconel 625-NbC materials?

13) There exist a lot of grammar errors. Should be proofread by professionals.
